# Emergence of ferroelectricity in a nonferroelectric monolayer

Wenhui Li[1,2,7], Xuanlin Zhang[3,7], Jia Yang[2,4,7], Song Zhou[1,2], Chuangye Song [1,5], Peng Cheng [1,2], Yi-Qi Zhang[1,2], Baojie Feng [1,2], Zhenxing Wang [2,4], Yunhao Lu [3,6] ✉, Kehui Wu [1,2,5] ✉ & Lan Chen [1,2,5] ✉

Ferroelectricity in ultrathin two-dimensional (2D) materials has attracted broad interest due to potential applications in nonvolatile memory, nanoelectronics and optoelectronics. However, ferroelectricity is barely explored in materials with native centro or mirror symmetry, especially in the 2D limit. Here, we report the first experimental realization of room-temperature ferroelectricity in van der Waals layered GaSe down to monolayer with mirror symmetric structures, which exhibits strong intercorrelated out-of-plane and in-plane electric polarization. The origin of ferroelectricity in GaSe comes from intralayer sliding of the Se atomic sublayers, which breaks the local structural mirror symmetry and forms dipole moment alignment. Ferroelectric switching is demonstrated in nano devices fabricated with GaSe nanoflakes, which exhibit exotic nonvolatile memory behavior with a high channel current on/off ratio. Our work reveals that intralayer sliding is a new approach to generate ferroelectricity within mirror symmetric monolayer, and offers great opportunity for novel nonvolatile memory devices and optoelectronics applications.

Ferroelectricity is a collective polarization effect arising from the spontaneous alignment of electric dipoles, in which the polarization states can be switched by an external electric field[1]. The ferroelectric field-effect transistor (FeFET) is considered as a good candidate for next-generation non-volatile memory because of its non-destructive read and fast repeatable write characteristics, and its non-volatile functionality is realized by the bistable ferroelectric polarization in the ferroelectric layer[2]. Conventional ferroelectricity occurs in non-centrosymmetric materials like three-dimensional (3D) perovskite oxides, such as $BaTiO_3$, $PbTiO_3$, and $BiFeO_3$[3,4]. In recent years, two-dimensional (2D) ferroelectricity has attracted considerable research interest[5], and a few 2D van der Waals (vdW) layered materials with intrinsic ferroelectricity have been experimentally verified, such as $CuInP_2S_6$[6,7], IV–VI group compounds ($SnTe$[8], $SnSe$[9], and $SnS$[10,11]), $In_2Se_3$[12–17], $Bi_2O_2Se$[18]. In these 2D vdW ferroelectrics, the spontaneous

electric polarization originates from the non-centrosymmetric structure in the monolayer, and such intrinsic 2D vdW ferroelectrics are rather rare owing to the strict constraints in lattice symmetry. On the other hand, due to the weak vdW interaction between neighboring layers of 2D layered materials, it is relatively easy to slide the layers, which breaks centrosymmetry and leads to the emergence of vertical polarization[19]. This type of vertical ferroelectricity is so-called inter-layer-sliding ferroelectricity, which has been experimentally demonstrated in bilayer transition metal dichalcogenides (TMDs, $WTe_2$[20–22], $WSe_2$[23], and $MoS_2/WS_2$[24]), bilayer $h$BN[25,26] and 1T'-$ReS_2$[27]. However, the interlayer-sliding ferroelectricity in bilayer or multilayer systems generally exhibits out-of-plane (OOP) polarization but lacks in-plane (IP) polarization, and the polarization field is much smaller than intrinsic ferroelectrics in monolayers with non-centrosymmetric structure. Efforts to explore ferroelectricity in 2D materials with various

[1]Institute of Physics, Chinese Academy of Sciences, Beijing 100190, China. [2]School of Physical Sciences, University of Chinese Academy of Sciences, Beijing 100190, China. [3]State Key Laboratory of Silicon Materials, School of Materials Science and Engineering, Zhejiang University, Hangzhou 310027, China. [4]National Center for Nanoscience and Technology, Chinese Academy of Sciences, Beijing 100190, China. [5]Songshan Lake Materials Laboratory, Dongguan, Guangdong 523808, China. [6]Zhejiang Province Key Laboratory of Quantum Technology and Device, School of Physics, Zhejiang University, Hangzhou 310027, China. [7]These authors contributed equally: Wenhui Li, Xuanlin Zhang, Jia Yang. ✉e-mail: luyh@zju.edu.cn; khwu@iphy.ac.cn; lchen@iphy.ac.cn

symmetric structures not only expand the ferroelectric materials family but also shed light on the mechanism of 2D ferroelectricity.

In recent years, GaSe has attracted much attention because it possesses many unique physical properties, such as topological insulator phase transition[28], Lifshitz transition induced by sombrero-shape valence band[29], tunable parabolic to ring-shaped valence band[30], tunable ferromagnetism and half-metallicity[31]. GaSe belongs to the group of III−VI post-transition metal chalcogenides compound semiconductors. The 2D vdW polytypes usually come from layers with identical structures but different interlayer stacking sequences[32]. The bulk GaSe consists of 2D vdW-bonding stacked layers with each Se-Ga-Ga-Se atomic sublayer (Fig. 1a) and mainly has four different polytypes, named β-(2H), ε-(2 R), γ-(3 R), and δ-(4H) (Supplementary Fig. 1), corresponding to AA′AA′..., ABAB..., ABCA... and AA′B′B...stacking sequences between the adjacent layers, respectively. Obviously, all the GaSe polytypes have the same primitive unit cells with a mirror symmetric structure in the monolayer. According to classical theory, ferroelectricity should be absent in GaSe monolayers, although vertical polarization was theoretically predicted to occur in bilayer or multilayer GaSe via interlayer sliding[19]. To date, there is no experimental report on ferroelectricity in monolayer or even multilayer GaSe.

In this work, few-layer γ-GaSe were prepared by both mechanical exfoliation and the molecular beam epitaxy (MBE) technique. The crystal structure of GaSe was verified through combined measurements of X-ray photoelectron spectroscopy (XPS), X-ray diffraction (XRD), Raman spectroscopy, and second-harmonic generation (SHG). The intercorrelated OOP and IP ferroelectricity in multilayer or even down to monolayer GaSe at room temperature is revealed by piezo-response force microscopy (PFM). High-angle annular dark-field scanning transmission

electron microscopy (HAADF-STEM) and density functional theory (DFT) calculations reveal the intralayer sliding of the Se sublayer in monolayer GaSe, which breaks the mirror symmetry of monolayer structure and induces alignment of dipole moments, leading to ferroelectricity. Furthermore, we fabricated a FeFET device based on GaSe nanoflakes as channel materials, which exhibits voltage-tunable ferroelectricity switching behavior with a high channel current on/off ratio ($10^3$) and further indicates both IP and OOP polarization in GaSe. The FeFET device exhibits reversible polarization and can be repeatedly rewritten over thousands of cycles without losing the nonvolatile ferroelectric memory characteristics. Our work suggests that significant ferroelectricity can exist in 2D materials with intralayer sliding and demonstrates the promising application of 2D ferroelectric-based memory devices.

## Results

### Characterization of γ-GaSe

The γ-GaSe has a hexagonal symmetric crystal structure with $a = b = 3.75$ Å, $c = 24$ Å, $\alpha = \beta = 90°$, and $\gamma = 120°$, belonging to the $R3m$ space group ($C_{3v}^5$)[33–35]. Few-layer GaSe nanoflakes were mechanically exfoliated from bulk using polydimethylsiloxane (PDMS) tape and then transferred onto $SiO_2$/Si substrate. The Raman spectrum measured (Fig. 1b) on GaSe nanoflakes presents seven Raman active phonon modes at 19.4 cm$^{-1}$ ($E_{2g}^2$), 59.1 cm$^{-1}$ ($E_{1g}^1$), 134.3 cm$^{-1}$ ($A_{1g}^1$), 212.9 cm$^{-1}$ ($E_{2g}^2$), 235 cm$^{-1}$ ($A_2''$(TO)), 247.5 cm$^{-1}$ ($A_2''$(LO)), 307.8 cm$^{-1}$ ($A_{1g}^2$), where TO and LO stand for the transverse optical and longitudinal optical phonons, respectively. The major difference between γ and other phases is that the enhancement of the features at approximately 235 cm$^{-1}$ and

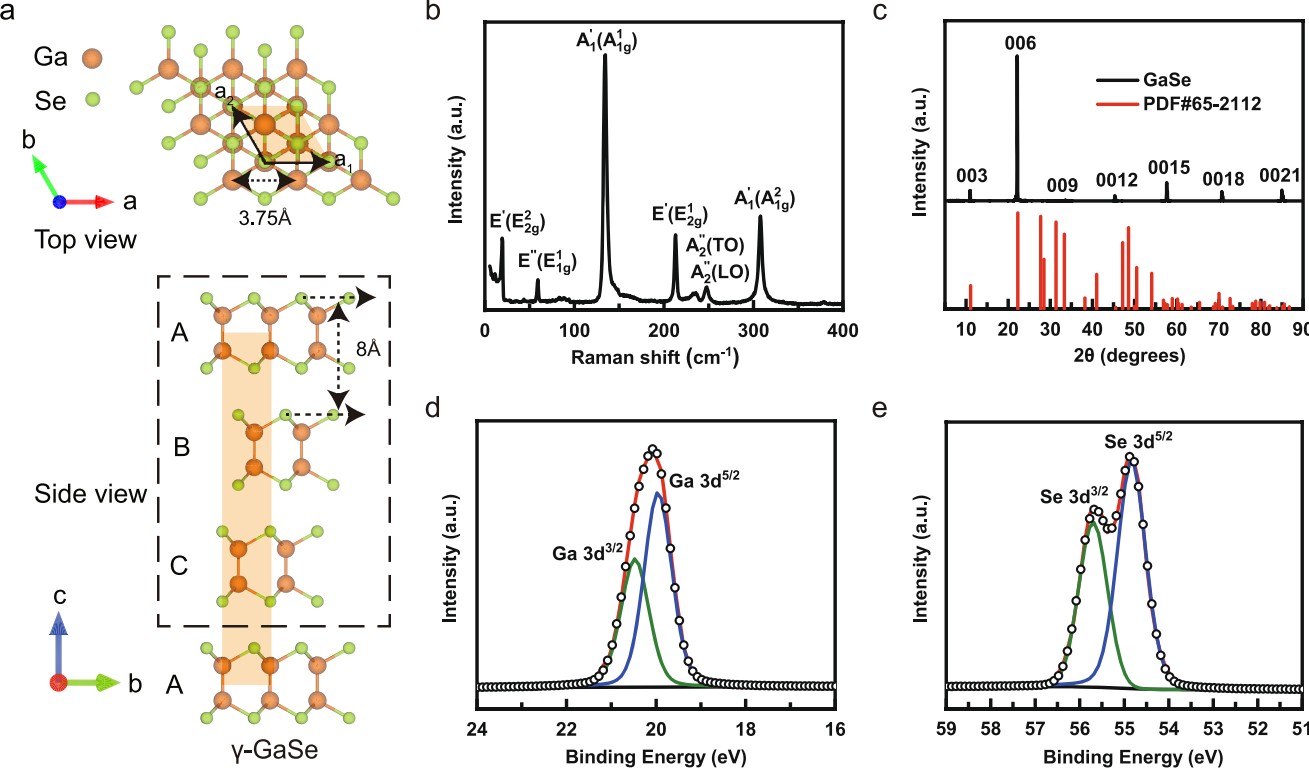

**Fig. 1 | The crystal structure and characterization of γ-GaSe. a** Schematic top and side view of the γ-GaSe structure. The orange and green spheres refer to the Ga atoms and Se atoms, respectively. The orange rhombus shadow indicates a primitive cell in the top view. Each quadruple layer (QL) contains four atomic layers in the order of Se-Ga-Ga-Se layers, and three QLs form a unit cell (dashed rectangle). **b** Raman spectrum of γ-GaSe nanoflake. **c** XRD pattern of the γ-GaSe crystal (upper panel) and the corresponding XRD standard card (lower panel, PDF#65-2112). **d, e** XPS spectrum of Ga 3*d* and Se 3*d* core orbital peaks took on the γ-GaSe nanoflake, respectively. The small circles represent the fitting result of the experimental data (red curves), decomposed into doublet peaks (olive and blue curves).

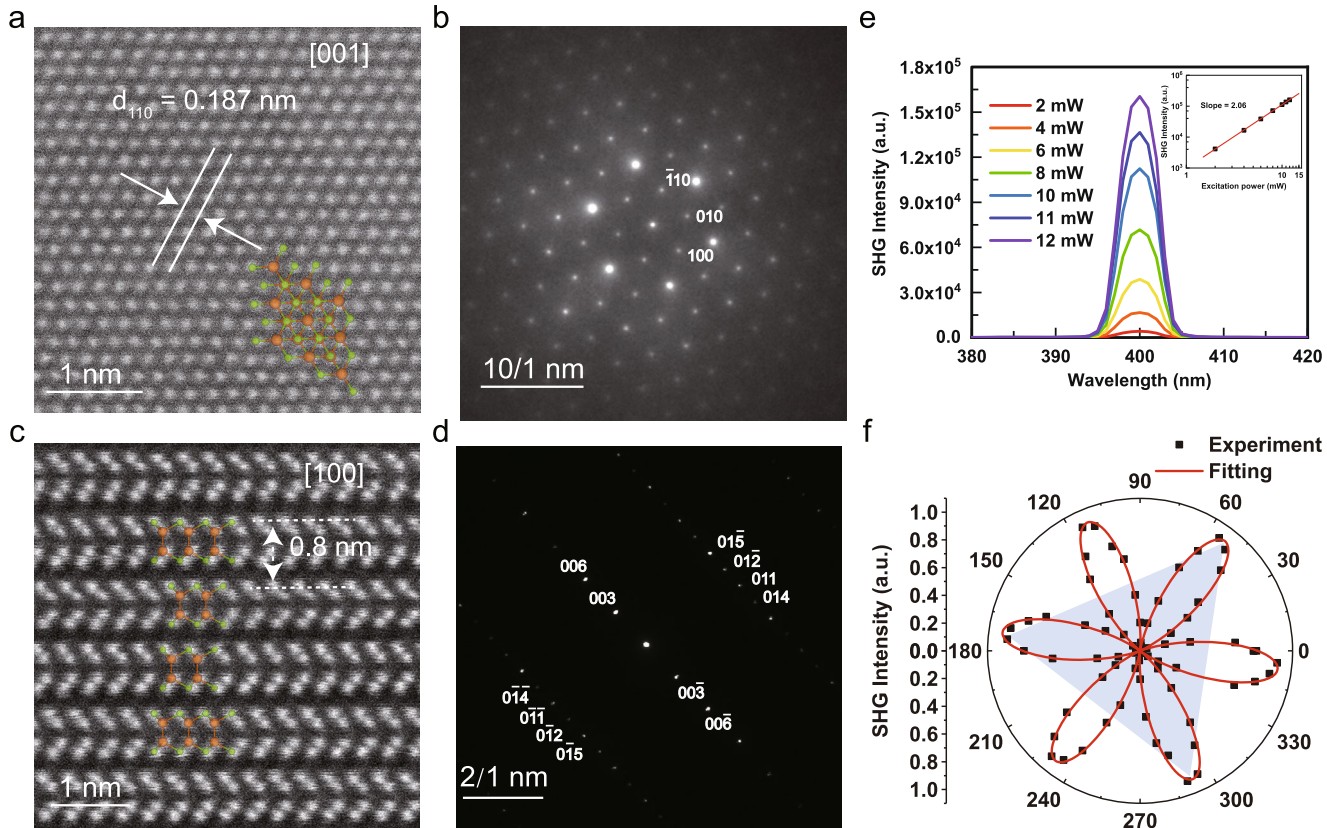

**Fig. 2 | HAADF-STEM images and SHG signals of γ-GaSe. a, b** The planar-view HAADF-STEM image and corresponding SAED pattern of γ-GaSe along the [001] zone axis, respectively. **c, d** The cross-section HAADF-STEM image and corresponding SAED pattern of γ-GaSe along the [100] zone axis, respectively. **e** SHG spectra generation from γ-GaSe nanoflake with different excitation powers. Inset: the SHG intensity exhibits a power-law relationship with excitation power, and the coefficient is fitted to 2.06. **f** Polarization angle $\theta$ dependent SHG intensity. The red line is the fitted curve of the scattered experimental data.

247.5 cm$^{-1}$ is due to the change of $A_2''$ mode activity in γ-GaSe[36–39]. Therefore, we assign our GaSe sample to the predominantly γ phase.

The crystalline structure of GaSe nanoflakes was further confirmed by XRD, as shown in Fig. 1c. The diffraction peaks correspond to the (003), (006), (009), (0012), (0015), (0018), and (0021) planes with a series of equal space, agreeing with the data in XRD standard card (PDF no. 65-2112) which can be indexed as the rhombohedral structured γ-GaSe phase belonging to the *R3m* space group[40,41]. The XPS spectrum of GaSe nanoflake reveals binding energies of Ga-3$d_{5/2}$ and Ga-3$d_{3/2}$ at 19.96 and 20.47 eV, and Se-3$d_{5/2}$ and Se-3$d_{3/2}$ at 54.83 and 55.7 eV, respectively (Fig. 1d, e), which is consistent with previous reports[42,43]. The stoichiometry of GaSe can be quantified by the integrated areas of fitted peaks, which give a Ga:Se ratio of ~1.05. Furthermore, the energy-dispersive spectroscopy (EDS) of GaSe nanoflakes suggests the atomic ratio of Ga:Se chemical composition is approximately 1:1 (Supplementary Fig. 2), confirming the composition purity of the GaSe nanoflakes.

**HAADF-STEM and SHG measurement**

To further investigate the structural details at the atomic scale, the few-layer GaSe sample exfoliated from bulk GaSe was transferred onto a grid and investigated by STEM. The low-magnification STEM images of GaSe nanoflake are shown in Supplementary Fig. 3. Figure 2a displays the planar-view HAADF-STEM image of GaSe taken along the [001] axis; the d$_{110}$ spacing between the neighboring (110) planes is 0.187 nm, reflecting the IP lattice constant is approximately 0.374 nm. The corresponding electron diffraction (SAED) pattern shows an obviously six-fold symmetry (Fig. 2b). From the cross-section HAADF-STEM image of

GaSe nanoflakes (Fig. 2c) along the [100] zone axis, the stacking sequence of GaSe can be identified as ABCABC..., confirming the assignment of our sample to the γ phase. The d spacing between the neighboring (003) planes of γ-GaSe phase is approximately 0.8 nm. However, since the atomic number of Se (Z = 34) is close to Ga (Z = 32), the integrated intensity of the two atomics is similar in HAADF-STEM images, making it difficult to distinguish between them. Nevertheless, the IP and OOP crystalline lattices are clearly distinguished and perfectly agree with the γ-GaSe polytype. Additionally, the corresponding SAED pattern indicates a rhombohedral structure (Fig. 2d), and the closely spaced reflection spots correspond to the (003) (006) (01$\bar{2}$) (01$\bar{5}$) (011) (014) planes of γ-GaSe phase, respectively.

SHG is a powerful technique to investigate the symmetry properties of the crystal structure. We detected the nonlinear emission spectrum on GaSe nanoflakes using a femtosecond OPO laser pulse with an excitation wavelength of 800 nm. The experimental setup for the SHG measurement schematic is shown in Supplementary Fig. 4. As shown in Fig. 2e, the typical nonlinear spectrum indicates a prominent peak centered at 400 nm, which is exactly half of the excitation wavelength. This frequency doubling of the excitation frequency indicates that our GaSe sample has a non-centrosymmetric structure. Moreover, increasing the excitation power from 2 to 12 mW also enhances the SHG intensity. The dependence of SHG intensity on the excitation power exhibits a power-law relationship. The coefficient is fitted to be 2.06 (inset of Fig. 2e), which accords with the electric dipole theory, confirming that the peak at 400 nm is an SHG response. Then, we investigated the structural symmetry of GaSe through the polarization angle $\theta$ dependent SHG intensity. The SHG

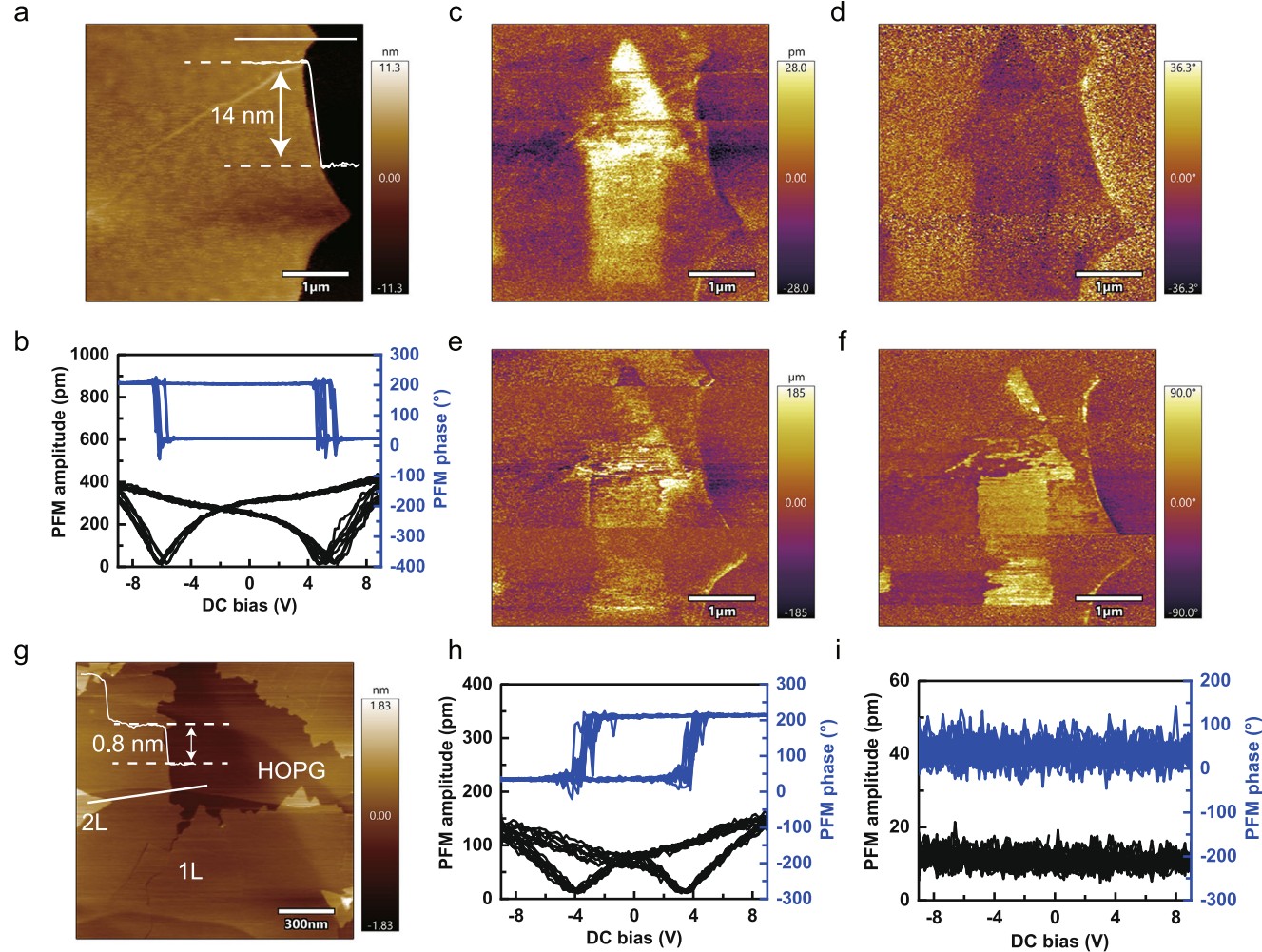

**Fig. 3 | Ferroelectric polarization switching by PFM on γ-GaSe nanoflakes.**
**a** Atomic force microscopy (AFM) topography of an exfoliated γ-GaSe nanoflake
with a 14 nm thickness. **b** The local SS-PFM performed on a GaSe nanoflake. **c–f** The
PFM OOP amplitude (**c**), OOP phase (**d**), IP amplitude (**e**), and IP phase (**f**) images of

the GaSe nanoflake after writing an arrow symbol pattern with opposite DC bias
(±10 V). **g** AFM topography of monolayer GaSe grown on HOPG substrate. **h, i** The
local SS-PFM performed in monolayer GaSe and HOPG substrate, respectively.

intensity dependent on the polarization angle $\theta$ exhibits a clear
six-fold symmetrical petal shape (Fig. 2f), which can be well fitted
by the formula, $I = I_0 \cos^2(3\theta + \theta_0)$, where $I$, $I_0$, $\theta$ and $\theta_0$ represent
the SHG intensity, maximum SHG intensity, rotation angle, and
initial crystallographic orientation, respectively. The SHG signal
indicates our GaSe sample possesses an inversion symmetry-
broken structure, suggesting the possibility of ferroelectric
polarization.

**Ferroelectricity in γ-GaSe**
To confirm whether ferroelectricity exists in GaSe, we performed
PFM measurements with local switching spectroscopy (SS) on
exfoliated γ-GaSe nanoflake with a 14 nm thickness on an Au-
coated SiO$_2$/Si substrate (Fig. 3a). The local PFM amplitude and
phase hysteretic loops were observed in GaSe nanoflakes at room
temperature, as shown in Fig. 3b, which exhibit typical ferro-
electric butterfly shaped amplitude-voltage loops and hysteresis
loops with the distinct 180° phase switching, confirming the OOP
ferroelectric polarization in γ-GaSe nanoflakes. Both the butterfly-
shaped loops and hysteresis loops remained consistent after 9
cycles, indicating that the ferroelectricity of γ-GaSe is quite
robust. The coercive voltage is approximately 6.0 V, which is
larger than In$_2$Se$_3$ (1.5 V)[14], SnS (2.0 V)[10], and CuInP$_2$S$_6$ (4.0 V)[7],
implying a higher energy barrier between the two opposite

polarization states. Another characteristic of ferroelectrics is
domain engineering by an electric field. We manipulated the
polarization reversal in the ferroelectric domain through PFM
electrical "read and write" operations. Figure 3c–f shows the PFM
OOP and IP amplitude and phase images of GaSe nanoflakes after
writing an arrow symbol pattern with opposite DC bias (+10 V and
−10 V). Moreover, the IP amplitude and phase were read syn-
chronously with the OOP amplitude and phase, indicating that the
IP ferroelectricity is intercorrelated with the OOP ferroelectricity
in GaSe. In addition, we discussed the reasons why ferroelectric
domains could not flip the 180° phase (Supplementary Note 1).

We also investigated the ferroelectricity of monolayer GaSe
on Au-coated SiO$_2$/Si substrates (Supplementary Fig. 5). The fer-
roelectric butterfly-shaped amplitude-voltage loops and hyster-
esis loops with the 180° phase reversed were also clearly
observed. For the polarization of ferroelectric domains inversion
by "read and write" operations, the OOP and IP polarization sig-
nals were faintly displayed. Therefore, the OOP and IP ferro-
electric polarization could still be maintained in GaSe with
thickness down to monolayer. It is worth noting that similar
hysteretic loops may also emerge in some non-ferroelectric
materials due to the non-local electrostatic or ion migration
effect[44,45]. To exclude these possibilities, we grew monolayer GaSe
on HOPG substrate by MBE to perform PFM experiments, as

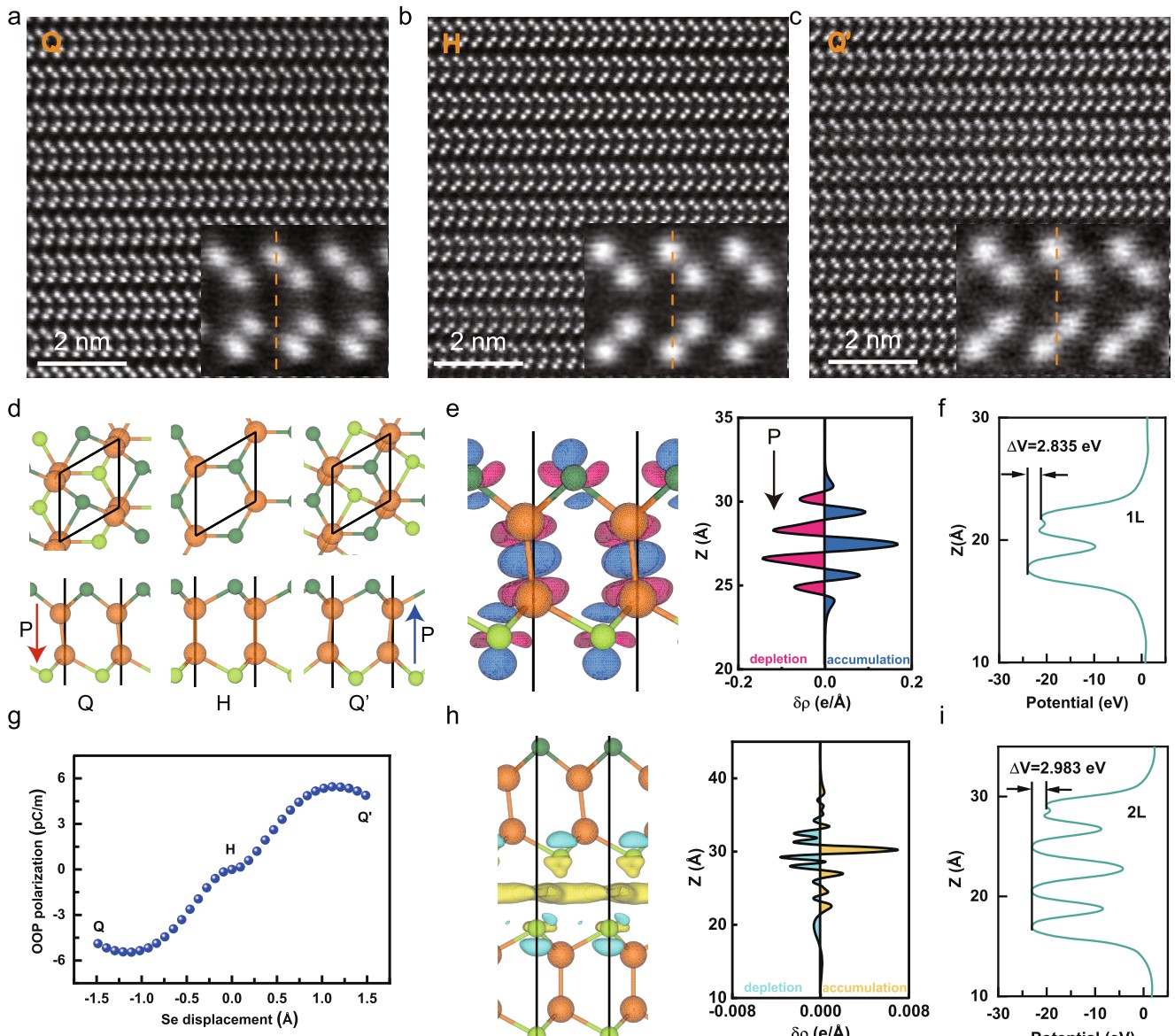

**Fig. 4 | Theoretical calculations on ferroelectricity in GaSe. a–c** STEM images of distorted Q phase (**a**), non-distorted H phase (**b**), and distorted Q′ phase (**c**) of GaSe. The distorted Q′ phase is generated by intralayer sliding of the Q phase after the applied electric field polarization. The insets are enlarged images, respectively. **d** The geometric structure of the Q, H, and Q′ phases GaSe monolayer. The orange, dark green, and light green balls denote Ga atoms and Se atoms of the upper sublayer and bottom sublayer. The red and blue arrows represent the direction of polarization. **e** The differential charge density between the upper sublayer and bottom sublayer in real space (isosurface 0.0025 e/Bohr³, left panel) and corresponding planar averaged differential charge density (right panel) in the GaSe monolayer. The blue and pink colors represent electron accumulation and depletion regions, respectively. **f** The local potential of the Q phase along the z direction in monolayer. **g** The variation of OOP (blue dots) polarization is described by Se atom displacement, and the zero displacements are defined as the H phase. **h** The differential charge density between the topmost Q phase and the remaining layers in real space (isosurface 0.0001 e/Bohr³, left panel) and corresponding planar averaged differential charge density (right panel) in bilayer GaSe. The light blue and yellow colors represent electron accumulation and depletion regions, respectively. **i** The local potential of the Q phase along the z direction in the bilayer.

shown in Fig. 3g. The conducting HOPG substrate can provide better ground contact and uniform electric field to avoid non-uniform field distribution or ion migration effect. The local SS in Fig. 3h indicates similar ferroelectric butterfly-shaped loops and hysteresis loops, while there is no hysteresis curve on the bare HOPG substrate (Fig. 3i). Therefore, we can conclude that OOP and IP ferroelectricity remains in GaSe even down to monolayer.

## Theoretical calculations

Since ferroelectricity can exist in monolayer GaSe, we believe that the ferroelectricity should not originate from interlayer sliding in the γ-GaSe phase. We observed two distinct structures in different regions of the GaSe sample using STEM: the distorted Q phase (Fig. 4a) and the non-distorted H phase (Fig. 4b). As shown in the inset of Fig. 4a, we found the Se atoms in the upper sublayer slightly shift away from the IP positions of the Se atoms in the bottom layer, and there is also slight translation between GaSe sublayers. Such local structure distortion can break the inversion symmetry in the monolayer and result in polarization. In addition, we polarized the GaSe sample at −50 V for 1 min and performed an ex situ STEM test, then observed the distorted Q′ phase (Fig. 4c), showing an opposite distortion direction to the Q phase. We speculate that the ferroelectric Q′ phase is generated by the intralayer sliding and polarization flip of the Q phase. Next, we performed theoretical calculations to verify this hypothesis. Based on the

STEM images of distorted GaSe before and after ferroelectric switching, we construct the ferroelectric Q and Q' phase in DFT calculations, as shown in Fig. 4d. The distortion of the Q phase breaks the mirror symmetry in the prototype GaSe monolayer. Thus inducing an unbalanced charge distribution between the sublayers. As shown in Fig. 4e, Ga atoms share electrons with each other to form covalent bonds. Thus there are electron accumulation regions between them and electron depletion regions around Ga atoms. Meanwhile, the ionic Ga–Se bonds in the sublayer dominate, and more electrons will stay around Se atoms, as shown in the blue regions. In the ferroelectric Q phase, 0.13 electrons are transferred from the bottom sublayer to the upper sublayer, as shown in Supplementary Fig. 6, resulting in downward OOP polarization of 4.89 pC/m. This is consistent with the potential difference of 2.835 eV generated between sublayers by the intralayer shift of Se atoms in Fig. 4f. However, the energy of the distorted Q phase is higher than the mirror-symmetric H phase by 0.68 eV as shown in Supplementary Fig. 7. Thus, we propose that distorted Q phase may be produced by the built-in electric or strain field. Under the electric field, a portion of upper sublayer Se atoms in the H phase move along [110] direction, climbing over a barrier of 0.72 eV/u.c. to the metastable Q phase. After the electric field is removed, the Q phase can be maintained due to the presence of a barrier. Under the reversal of the electric field, the upper layer Se atoms first move back to the H phase without OOP polarization. Then, a portion of Se atoms in the bottom sublayer move to the Q' phase, and the OOP polarization is reversed to 4.89 pC/m pointing upward (Fig. 4g). Under experimental conditions with extended sample thickness and p-type doping, the energy barrier will be further reduced (Supplementary Fig. 8), which can be overcome by applying a large coercive field (6 V). Overall, it is the built-in electric or strain field that facilitates the coexistence of ferroelectric Q and H phases in the original STEM images, and the Q phase can be reversed to the Q' phase under the electric field. To conclude, the ferroelectricity detected in GaSe monolayer by PFM is attributed to intralayer sliding. This is completely different from the moiré ferroelectricity discovered recently in bilayers or multilayers vdW materials, where the symmetry breaking and ferroelectricity are due to the interlayer sliding, and the monolayer still holds symmetry.

Next, we focused on multilayer GaSe films. Here, local distortions only occur in the topmost layer of γ-GaSe films, and the rest maintain the prototype GaSe structure without distortion. The OOP polarization of such bilayer GaSe increases to 5.47 pC/m, larger than that of the monolayer. Correspondingly, the potential difference increases by 2.983 eV compared to the monolayer as the OOP polarization increases in Fig. 4i. The OOP polarization increases with layers and reaches 6.19 pC/m at trilayer with a potential difference of 3.011 eV (Supplementary Fig. 9). Such enlargement of polarization compared to monolayer is mainly due to the additional charge transfer between the topmost distorted layer and neighboring prototype GaSe layers (Fig. 4h, Supplementary Fig. 10 and Table S1). This is also consistent with the experimental observation of stronger polarization in thicker films. It should be emphasized that the polarization of the distorted Q phase is an order of magnitude larger than that of the interlayer sliding bilayer prototype GaSe (0.39 pC/m). The intralayer sliding not only causes such a large polarization in multilayer films but also the polarization down to the monolayer limit. In contrast, interlayer sliding ferroelectricity alone cannot cause such a large polarization strength, nor can it induce polarization in monolayer[19].

## FeFET devices

Finally, to demonstrate the potential application of GaSe in nonvolatile memory, we fabricated bottom-gated FeFET devices based on ferroelectric polarization switching, as shown in Fig. 5a, b. In the FET device, a 15 nm-thick GaSe nanoflake acts as the tunneling layer, 300-nm-thick SiO$_2$ as the dielectric layer, and two parallel Au/Cr electrodes as the source and drain (Supplementary Fig. 11). We obtained a Kelvin probe

force microscopy (KPFM) image of the GaSe FET device (Fig. 5c), indicating GaSe has a 140 mV higher contact potential barrier than the gold electrode. The transfer characteristic curve $I_{DS}$–$V_{GS}$ (Fig. 5d) was collected as $V_{GS}$ swept from −80 V to 80 V and then reversibly swept from 80 V to −80 V. The current hysteresis loop characteristic obtained by gate-write and channel-read indicates that the intercorrelated lateral and vertical ferroelectric polarization in GaSe can be regulated by gating. The behavior of the transfer characteristic curve $I_{DS}$–$V_{GS}$ at different $V_{DS}$ (5 V, 10 V) is consistent, indicating that the transition trend of the current is related to the ferroelectric polarization reversely induced by the applied electric field switching. The gate-tunable ferroelectric hysteresis characteristic results in low resistance state (LRS) and high resistance state (HRS) switching. The effect of trap states can be excluded from the unchanged hysteresis loops windows at different sweep rates (Supplementary Fig. 12) since the hysteresis loops windows induced by the trap states usually become larger when slowing down the voltage sweep rate[46,47].

The hysteresis loop of the transfer curve $I_{DS}$–$V_{GS}$ is obtained by switching the gate electric field in the vertical direction, which mainly reflects the existence of OOP ferroelectric polarization in GaSe. Similarly, the hysteresis loop of the output curves $I_{DS}$–$V_{DS}$ is obtained by switching the source-drain current in the lateral direction, which reflects the IP ferroelectric polarization characteristic (Fig. 5e). The hysteresis windows of the output curves $I_{DS}$–$V_{DS}$ incrementally expand as the maximal sweeping $V_{DS}$ increases, which is the characteristic behavior of ferroelectric memristors[48]. As the voltage sweeps from 0 V to −80 V (sweep I), the device enters LRS. While sweeping from the −80 V to 0 V (sweep II), it switches to HRS. Likewise, as the voltage sweeps from 0 V to 80 V (sweep III), the device enters LRS. While sweeping from 80 V to 0 V (sweep IV), it switches to HRS. In this way, the device can achieve HRS-LRS switching memristor functionality. The channel current under negative bias is two orders of magnitude larger than that under positive bias, which may be attributed to the switch diode effect caused by the Schottky barrier of the FeFET device. Notably, a higher sweeping $V_{DS}$ generates a lower current, as shown in the positive $V_{DS}$ part in Fig. 5e, which should be due to the IP ferroelectric polarization. The channel currents and hysteresis windows distinctly increase as the applied gate voltages vary from 80 V to −80 V (Fig. 5f), suggesting the FeFET device possesses gate-tunable ability owing to the ferroelectric polarization switching in GaSe. The hysteresis curves are distinctly observed with gate-write and channel-read, indicating that not only the OOP ferroelectric dipoles but also IP ferroelectric dipoles can be reversed by applying a vertical electric field, suggesting that the IP and OOP ferroelectric dipoles are intercorrelated in GaSe. The hysteresis output curves $I_{DS}$–$V_{DS}$ exhibits a gate-tunable effect by applying gate voltage from 80 V to −80 V. The switchable diode effect can be regulated after pulse gate voltage bias poling. The output hysteresis curves change significantly, poled by applying a positive (80 V) and negative (−80 V) voltage pulse, respectively (Fig. 5g), which may stem from the asymmetric modulation of the Schottky barrier by ferroelectric polarization switching[49]. In addition, we discuss the resultant band alignments of the GaSe FET device (Supplementary Fig. 13).

To test the repeatability and reversibility of the lateral resistance switching of the GaSe FET memristor, we repeatedly pole the IP resistance switching by applying a periodic pulse channel bias (Supplementary Fig. 14a). The resistance switching exhibits a high LRS/HRS ratio of $10^3$ and superior reversibility over 3500 cycles by applying channel bias (Fig. 5h). Ferroelectric memristor can also regulate the resistance switching of source and drain by applying the gate voltage, which is very important for practical application. The lateral GaSe ferroelectric memristor based on IP polarization exhibits gate tunability by applying periodic pulse gate bias (Supplementary Fig. 14b).

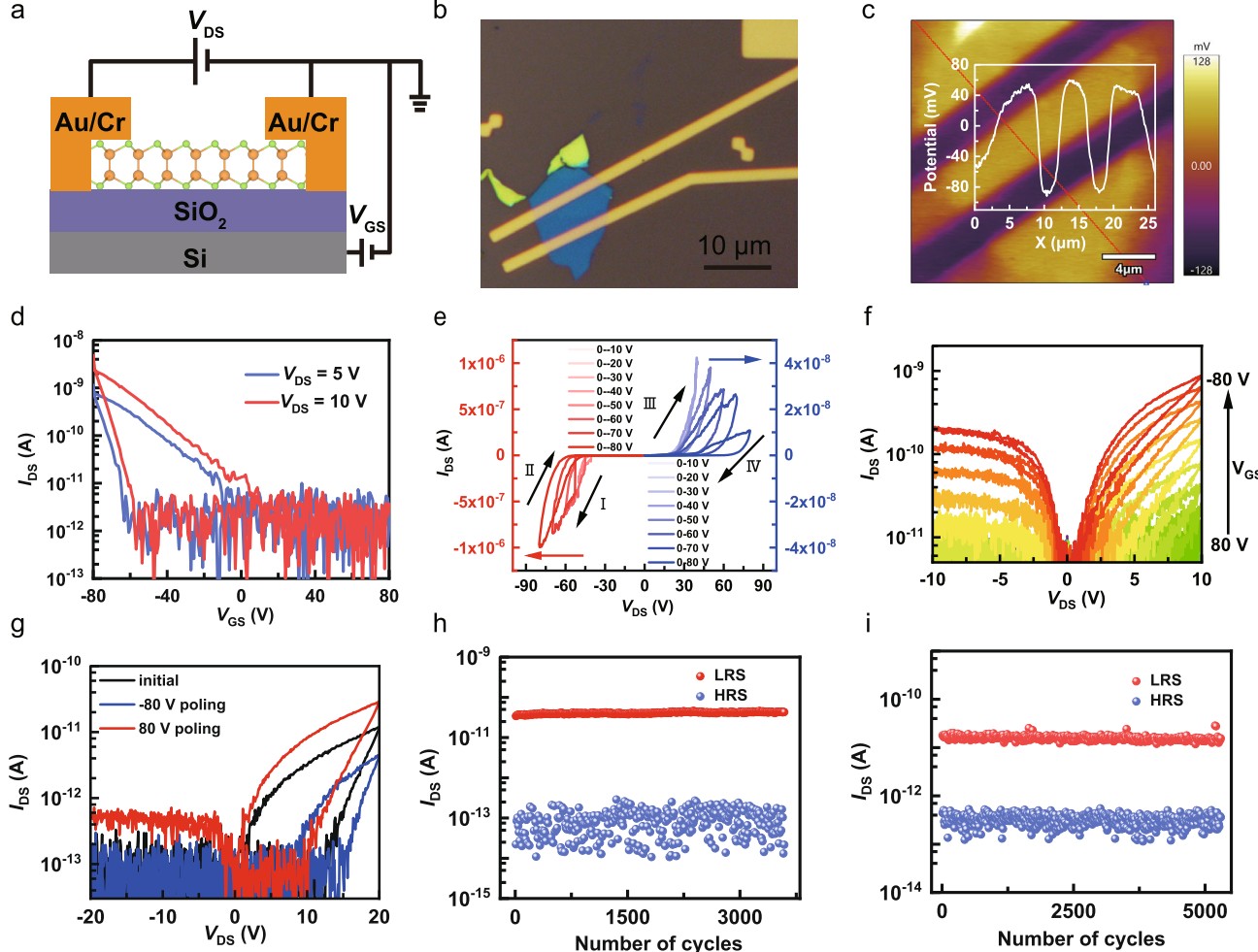

**Fig. 5 | Nonvolatile memory applications of GaSe FeFET devices. a, b** Schematic illustration and optical microscopy image of the devices, respectively. **c** The KPFM image of the GaSe FeFET device. **d** The transfer characteristic curve $I_{DS}$–$V_{GS}$ of GaSe nanoflake at different $V_{DS}$ (5, 10 V). **e** The hysteresis loop of the output curves $I_{DS}$–$V_{DS}$ at different sweeping maximal $V_{DS}$. **f** The hysteresis output curves $I_{DS}$–$V_{DS}$ exhibits a gate-tunable effect by applying gate voltage from 80 V to −80 V. **g** The switchable diode effect is regulated after pulse gate voltage bias poling (±80 V). **h, i** The resistance switching of the GaSe FeFET memristor by applying periodic pulse channel bias or gate voltage.

The IP resistance switching exhibits a considerable LRS/HRS ratio of $10^2$ and superior reversibility over 5000 cycles by applying the gate voltage due to the OOP ferroelectric polarization reversal (Fig. 5i). The retention performance of HRS for over 100 s was also carried out and shown in Supplementary Fig. 15. The HRS was stable for over 100 s and showed almost no current change during this period. Therefore, we demonstrated that intercorrelated IP and OOP ferroelectric polarization in GaSe can be reversed by applying channel or gate bias, and ferroelectric resistance switching exhibits good repeatability.

## Discussion

In summary, based on experimental characterizations and theoretical calculations, we have unambiguously demonstrated the existence of 2D ferroelectricity in GaSe down to monolayer. The emergence of IP and OOP ferroelectricity in the monolayer mirror symmetric structure originates from the intralayer sliding of the Se atomic sublayer. The strong polarization strength also indicates that intralayer sliding should be the main mechanism for ferroelectricity in multilayer GaSe. We have demonstrated that IP and OOP ferroelectric polarization in GaSe are intercorrelated and can be reversed by applying channel or gate bias. Ferroelectric resistive switching in GaSe FET memristors exhibits a considerable LRS/HRS ratio and strong reversibility. Our work not only expands the 2D ferroelectric family from the basic physical mechanism but also provides a new option for practical applications in ferroelectric nonvolatile devices.

## Methods

### Sample preparation
Bulk GaSe was purchased from Nanjing MKNANO Tech. Co., Ltd. (www.mukenano.com). GaSe nanoflakes were mechanically exfoliated from bulk GaSe using PDMS tape and then transferred onto SiO₂/Si substrates of evaporated gold for PFM measurement. In addition, we also grew monolayer GaSe on HOPG substrate by MBE. The optical absorption spectrum and photoluminescence (PL) of GaSe nanoflake are shown in Supplementary Fig. 16. The oxidation process of GaSe was investigated in Supplementary Note 2.

### Device fabrication and electrical measurement
GaSe nanoflakes were transferred onto 300 nm SiO₂/Si substrates, then parallel Au/Cr electrodes (80/8 nm) were deposited by an electron beam evaporative deposition process. Electrical properties were tested by a four-probe station (Lakeshore, TTP4) equipped with a Keysight B1500A semiconductor analyzer. Here, two modules are primarily applied: the DC voltage was performed on the B1517A module, and the pulse voltage emerged by the B1530A waveform generator module.

## Raman and PL spectroscopy characterization

Raman spectroscopy was carried out by using a LabRAM HR Evolution spectroscope with a 532 nm laser excitation. The low-frequency Raman phonon below 50 cm$^{-1}$ was collected by ultralow wave number mode. PL spectroscopy was collected using a 325 nm laser excitation.

## XRD measurement

The X-ray diffraction measurements were performed by X-ray diffractometer (Rigaku Ultima IV) with Cu K$_\alpha$ radiation, and the corresponding wavelength is 1.5406 Å in the scanning range (2$\theta$) 5°–90°.

## XPS spectra

XPS spectra were carried out by using Thermo Scientific ESCALAB 250X with a micro monochromatic Al-K$_\alpha$ X-ray source, the sampling area can be reduced to 200–300 μm, and the energy resolution can reach up to 0.45 eV.

## STEM characterization

The cross-sectional STEM specimens were prepared using a standard focused ion beam lift-off process. The atomic-resolution HAADF-STEM images and EDS mapping were performed on an ARM-200F (JEOL) high-resolution STEM operated at 200 kV, and a CEOS Cs corrector (CEOS GmbH) corrector is equipped to deal with the spherical aberration of the objective lens formed by the probe.

## Scanning probe microscopy measurement

AFM, KPFM, and PFM measurements were performed on GaSe nanoflakes by a commercial scanning probe microscope (Asylum Cypher ES AFM) at room temperature under ambient conditions. The local SS-PFM was collected in dual AC resonance tracking (DART) mode by superimposing an AC signal on a series of DC triangular, sawtooth waveform voltages. The PFM phase and amplitude signals were recorded when the applied voltage state was off. The switching voltage range of the ferroelectric hysteresis loops and domain writing was ±10 V.

## SHG measurement

Optical SHG signals were collected on a laser scanning confocal microscopy (LSM)-fluorescence lifetime imaging (FLIM) combined system (Nikon-ARsiMP-LSM-Kit-Legend Elite-USX). A Ti: sapphire femtosecond laser is used as the excitation source to generate the ultrafast pulsed light, in which the output wavelength is 800 nm, the repetition frequency is 80 MHz, and the pulse light width is less than 140 fs. The SHG signals were performed in the reflection geometry. The polarization direction of the incident light was regulated by rotating the $\lambda/2$ wave plate. P- and S-polarized components of the generated second harmonic field ($E_{2\omega}$) can be decomposed by a polarizing beam splitter. A high-pass filter is applied to remove low-frequency signals from the excitation light. The optical SHG signals were detected by a photon multiplier tube.

## Absorption spectrum measurement

The absorption spectrum was performed on an ultraviolet-visible near infrared spectrophotometer (Lambda 1050+, Perkin Elmer) with an excitation wavelength of 220–1100 nm.

## DFT calculation

First-principles calculations based on density functional theory were performed via the Vienna ab-initio simulation package[50] using the generalized gradient approximation[51] with the projected augmented wave method[52]. Perdew-Burke-Ernzerhof function was adopted for exchange-correlation functional. The energy cutoff for the plane wave basis expansion was set to 400 eV. The Becke–Jonson damping method (DFT-D3)[53] was employed to incorporate the effects of van der Waals interactions. A Γ-centered $k$-point grid of $12 \times 12 \times 1$ was applied, and the structures were fully relaxed until the energy and force converged to $10^{-5}$ eV and 0.01 eV/Å, respectively. A 50 Å vacuum layer was set to eliminate the interaction by the periodic boundary condition. The polarizations were calculated using the modern theory of polarization[54] as implemented in VASP. The charge on each atom was calculated by Mulliken population analysis[55,56] based on orbital basis code RESCU[57].

## Data availability

The data that support the findings of this paper are available from the corresponding authors upon request.

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

## Acknowledgements

This work was supported by the Ministry of Science and Technology (MOST) of China (Grant nos. 2018YFE0202700, 2021YFA1400502, and 2019YFE0112000), the National Natural Science Foundation of China (Grant nos. 12134019, 11825405, and 11974307), the Zhejiang Provincial Natural Science Foundation of China (LR21A040001), and the Chinese Academy of Sciences (Grant no. XDB30000000, YSBR-054). We gratefully acknowledge HZWTECH for providing computation facilities.

## Author contributions

W.L., X.Z., and J.Y. contributed equally to this work. L.C. and K.W. proposed and conceived this project. W.L., J.Y., S.Z., and C.S. contributed to the experiment. X.Z. and Y.L. provided theoretical support for the experiment. W.L., L.C., K.W., P.C., Y.Z., B.F., and Z.W. did the data analysis. W.L., L.C., Y.L., and K.W. wrote the paper with input and comments from all co-authors.

## Competing interests

The authors declare no competing interest.
