## [Peer Review File · Nature Communications]

REVIEWER COMMENTS

Reviewer #1 (Remarks to the Author):

In this manuscript, the authors have experimentally and theoretically demonstrated ferroelectricity in the monolayer of two-dimensional Gallium Selenide (GaSe) in its γ -(3R) phase. Moreover, the authors have experimentally realized a ferroelectric field effect transistor-based memristor, where the proof-of-concept has been presented for exploiting the inherent ferroelectricity of γ -GaSe in favour of a non-volatile memory device design. The proposed concept is novel, and the key findings are also promising for 2D non-volatile memory device design. However, in the reviewer's opinion, the authors must address some technical concerns regarding this work before the present manuscript can be addressed for publication in Nature Communication.

Typically, the following issues need to be addressed:

1. In the "Results and discussions" section under the "Characterization of γ -GaSe" sub-heading, authors need to cite appropriate experimental/ theoretical references for structural specifications in the following sentence
"The γ -GaSe has a hexagonal symmetric crystal structure with"
2. In the "Results and discussions" section under the "Theoretical calculations" sub-heading, authors need to specify the magnitude of the in-plane shifting between the top and bottom Se sub-layers.
3. In the "Results and discussions" section under the "Theoretical calculations" sub-heading, authors need to elaborate on the phrase "the reversal process" for general readers.
4. In the "Results and discussions" section under the "Theoretical calculations" sub-heading, authors are suggested to perform a Bader charge analysis and quantify the charge transfer from the top layer to the rest of the layers (bi-layer, tri-layer) of GaSe. Furthermore, the authors also suggested indicating the charge on each atom in monolayer GaSe in its three phases (Q, T-, and Q'-phase) from the Bader charge analysis to improve the analysis further.
5. In the "Results and discussions" section under the "Theoretical calculations" sub-heading, for general readers, the authors should briefly explain the origin of local electron depletion and accumulation regions in the different atomic positions of GaSe as depicted in the right panel of Fig. 4b. In this context, can the authors explain why a small kink is observed in the potential profiles (vertical direction) of the top sublayer of monolayer and few-layer γ -GaSe, as depicted in the Fig. 4c and 4f.
6. In the "Methods" section under the "DFT calculation" sub-heading, for general readers, the relaxed structural parameters (in-plane lattice constant, bond lengths, and bond angles) of monolayer γ -GaSe should be validated against previous experimental or theoretical reports.
7. In the "Methods" section under the "DFT calculation" sub-heading, the reviewer is concerned about the 20Å out-of-plane vacuum considered in this work, which might not be sufficient to eliminate the interactions from the out-of-plane periodic image completely. The authors need to justify this point.
8. The authors are requested to briefly discuss a few recent theoretical studies on GaSe in the "Introduction" section.

Reviewer #2 (Remarks to the Author):

In this manuscript, Li et al. reported signatures of room temperature ferroelectricity in GaSe flakes down to the monolayer limit. They claim the ferroelectricity comes from a new mechanism, i.e. intralayer sliding of the Se atomic sublayers. In the perspective of the reviewer, although the results appear interesting, the claim is still premature.

In several places (including in the title, Line 71-73 etc.), the authors refer monolayer or primitive unit cells of GaSe as “centrosymmetric”. In fact, only the primitive unit cell of β -GaSe (AA'AA'... stacking) is centrosymmetric, while the other three polytypes (including γ -GaSe) are not; monolayer GaSe (as monolayer MoS₂ in the 1H phase) is noncentrosymmetric. Maybe the authors mean nonferroelectric or nonpolar?

The authors also mentioned interlayer sliding ferroelectricity in bilayer TMDs, bilayer hBN and 1T' ReS₂. As we know, in van der Waals materials adjacent layers are bonded via weak van der Waals forces; within the layer, it's covalent bond (much stronger). So, one would expect intralayer sliding to be much more difficult than interlayer sliding. The authors should provide theoretical calculations of energies of the different phases (Q, T and Q'), and check the feasibility of intralayer sliding and ferroelectric switching.

Beside the above problems, I also have some technical questions.

1. Thin GaSe flakes are known to be air sensitive (Arora & Erbe, InfoMat 3, 662, 2021), they can degrade fast in ambient conditions, giving unrealizable PFM and transport signals. Have the authors tried to passivate the GaSe flakes, especially for monolayer GaSe? How stable are the measurements, for example over a few hours?
2. Following the above question, regarding transport measurements in Figure 5, one needs to be extremely careful in distinguishing ferroelectric hysteresis from charging effect that could possibly come from defects and impurities. Have the authors tried to sweep the gate/bias at different rates? How the HRS evolve as a function of time (retention time of the “ferroelectricity”)?
3. Voltage pulses and current plateaus in Supplementary Figure 7 do not look synchronized. Could the authors clarify the behaviors together with IV and transfer characteristics?
4. The authors claimed IP polarization signal is stronger than OOP polarization (Line 190) by comparing amplitudes of the lateral and vertical PFM. This is an unfair comparison since their responsivities could be rather different.

Responses to Reviewer #1

C: In this manuscript, the authors have experimentally and theoretically demonstrated ferroelectricity in the monolayer of two-dimensional Gallium Selenide (GaSe) in its γ - (3R) phase. Moreover, the authors have experimentally realized a ferroelectric field effect transistor-based memristor, where the proof-of-concept has been presented for exploiting the inherent ferroelectricity of γ -GaSe in favour of a non-volatile memory device design. The proposed concept is novel, and the key findings are also promising for 2D non-volatile memory device design. However, in the reviewer's opinion, the authors must address some technical concerns regarding this work before the present manuscript can be addressed for publication in Nature Communication.

R: We thank the Reviewer for the pertinent and constructive questions/comments on our manuscript. Based on the reviewer's suggestions, we have made careful revisions. More details can be seen in the following point-by-point responses.

C1. In the “Results and discussions” section under the “Characterization of γ -GaSe” sub-heading, authors need to cite appropriate experimental/ theoretical references for structural specifications in the following sentence

“The γ -GaSe has a hexagonal symmetric crystal structure with

R1: We are very grateful to the reviewer for the constructive suggestion, we have cited references [*Nanoscale* **12**, 8563-8573(2020); *Adv. Funct. Mater.* **31**, 2104965 (2021); *ACS Nano* **9**, 8078-8088 (2015)] related to the structure of GaSe in the revised manuscript.

C2. In the “Results and discussions” section under the “Theoretical calculations” sub-heading, authors need to specify the magnitude of the in-plane shifting between the top and bottom Se sub-layers.

R2: Thanks for the suggestion. We revised Fig. 4d, which is also shown as **Fig. R1**. The horizontal axis is substituted by the in-plane displacement of top and bottom Se atoms relative to the T phase (zero displacement).

Fig. R1. The variations of IP (blue dots) and OOP (red dots) polarizations with top and bottom Se displacements. The zero displacement is defined as the T phase.

C3. In the “Results and discussions” section under the “Theoretical calculations” sub-heading, authors need to elaborate on the phrase “the reversal process” for general readers.

R3: Thanks for the suggestion. To clarify “*the reversal process*” for general readers, here, we change the sentence “Furthermore, the IP and OOP polarizations are intercorrelated with each other during the reversal process as shown in Fig. 4d.” to “When the Se atoms of upper sublayer move along the [110] direction, GaSe undergoes a centrosymmetric T phase. Then, the Se atoms of the down sublayer similarly move along the [110] direction and climb over an energy barrier of 0.648 eV to enter the Q’ phase (Supplementary Fig. 5), leading to the simultaneous reversal of IP and OOP polarization. Under experimental conditions with an extended range of sample thickness and p-type doping, the energy barrier will be further reduced to 0.26 eV (Supplementary Fig. 6), which can be overcome by applying a moderate coercive field (6V). Furthermore, during the reversal process of polarization, the IP and OOP polarization are intercorrelated with each other as shown in Fig. 4d.”

C4. In the “Results and discussions” section under the “Theoretical calculations” sub-heading, authors are suggested to perform a Bader charge analysis and quantify the charge transfer from the top layer to the rest of the layers (bi-layer, tri-layer) of GaSe. Furthermore, the authors also suggested indicating the charge on each atom in monolayer GaSe in its three phases (Q, T-, and Q’-phase) from the Bader charge analysis to improve the analysis further.

R4: Thanks for the suggestions. Bader uses the so-called zero flux surfaces to divide atoms. A zero flux surface is a 2D surface on which the charge density is a minimum perpendicular to the surface. Typically, in molecular systems, the charge density reaches a minimum between atoms and this is a natural place to separate atoms from each other. However, for 2D system, the charge density minimum is not between atoms, and a natural place to separate atoms is not well-defined, especially for plane-wave basis. Thus, the *Bader charge analysis* is not quite accurate to quantify charge transfer here. Thus, we performed Mulliken population analysis [*J. Chem. Phys.* **23**,1833-1840 (1955); *J. Chem. Phys.* **23**, 1841-1846 (1955)], which is similar to Bader charge analysis, based on orbital basis code RESCU [*J. Comput. Phys.* **307**, 593-613 (2016)] to calculate the charge on each atom. As shown in **Fig. R2(a, c)**, there are 0.06 electrons transferred from bottom sublayer to upper layer in the Q phase, while the direction of electron transfer is reversed in the Q’ phase, which is consistent with the direction of OOP polarization in both Q and Q’ phase. In contrast, the charge distribution of the T phase is symmetric as in **Fig. R2b**, which indicates that the T phase is paraelectric.

Fig. R2. Charge transfer process in the ferroelectric polarization reversal. The Mulliken population on each atom in (a) Q, (b) T and (c) Q' phase. The curved arrows indicate the direction of charge transfer between GaSe sublayers. The straight arrows indicate the direction of OOP polarization.

To quantify the additional charge transfer between the top layer and the rest of the layers, we integrate the differential charge density in each part of multilayer GaSe, as shown in **Fig. R3** and **Table R1**. When increasing the number of GaSe layers, the charge transfer between the distorted Q phase and prototype γ -GaSe increases. Besides, the additional charge transfer is in the same direction as that in distorted Q phase monolayer, which corresponds to increased OOP polarization with the number of GaSe layers increasing. In conclusion, the OOP polarization of multilayer GaSe is contributed by both intralayer Se sliding and interlayer sliding.

Fig. R3. The differential charge density between the distorted Q phase and the rest of layers in real space (isosurface 0.0001 e/Bohr^3 , left panel) and corresponding planar averaged differential charge density (right panel) in (a) 2L, (b) 3L and (c) 4L GaSe films.

Table R1. The charge transfer calculated by integrating the differential charge between the topmost distorted Q phase and the rest of layers and the OOP polarization in 2L, 3L and 4L Q phase GaSe.

Layers	Charge transfer (e)	Polarization (pC/m)
2L	0.013	2.19
3L	0.022	3.13
4L	0.027	3.74

Revision: Based on the reviewer's comment, we have added **Table R1** as **Table S1** in the

revised Supplementary Information.

C5. In the “Results and discussions” section under the “Theoretical calculations” sub-heading, for general readers, the authors should briefly explain the origin of local electron depletion and accumulation regions in the different atomic positions of GaSe as depicted in the right panel of Fig. 4b. In this context, can the authors explain why a small kink is observed in the potential profiles (vertical direction) of the top sublayer of monolayer and few-layer γ -GaSe, as depicted in the Fig. 4c and 4f.

R5: Thanks for the comments. To briefly explain the origin of local electron depletion and accumulation regions in the different atomic positions of GaSe, we changed the sentence “As shown in Fig. 4(b, c), there is more electron accumulation in the upper sublayer than lower sublayer with a potential difference of 1.348 eV between them, resulting in OOP polarizations of 2.00 pC/m” to “As shown in Fig. 4b, Ga atoms share electrons with each other to form covalent bonds, thus there are electron accumulation regions between them and electron depletion regions around Ga atoms. Meanwhile, the ionic Ga-Se bonds in sublayer dominate and more electrons will stay around Se atoms, as shown in the blue regions. In the ferroelectric Q phase, 0.06 electrons are transferred from the bottom sublayer to the upper sublayer as shown in Supplementary Fig. 4, resulting in downward OOP polarization of 2.00 pC/m.”

To explain the origin of the kink, we have compared the local potentials of the Q and T phases. As shown in **Fig. R4**, the kink appears between the Ga atoms and sliding Se atoms. When Se atoms slide along [110] direction, one of the three equivalent Ga-Se bonds is elongated, while the other two also deviate from their stable bond lengths. As result, the Ga-Se bonds are weakened and the electrons between sliding Se and Ga atoms in Q phase will have higher energy, resulting in the kink.

Fig. R4. The local potential of the Q and T phase along the z direction in monolayer GaSe.

C6. In the “Methods” section under the “DFT calculation” sub-heading, for general readers, the relaxed structural parameters (in-plane lattice constant, bond lengths, and bond angles) of monolayer γ -GaSe should be validated against previous experimental or theoretical reports.

R6: Thanks for your comments. Here we list the relaxed lattice parameters, bond lengths and bond angles in **Table R2**. The structure parameters of monolayer γ -GaSe are comparable with previous calculations [*Phys. Rev. B* **95**, 115409 (2017)], and the

small difference is due to the different vdW correction (DFT-D3 method of Grimme) considered in our manuscript.

Table R2. The relaxed lattice parameters, bond lengths and bond angles of different monolayer GaSe phases.

	H	H (Ref.)	T	Q
Lattice parameters (Å)	3.77	3.82	3.78	3.77
Ga-Ga bond lengths (Å)	2.45	2.50	2.44	2.48
Ga-Se bond lengths (Å)	2.48	2.46	2.48	2.41
Ga-Se-Ga angles (°)	98.9	99.7	99.2	102.6

C7. In the “Methods” section under the “DFT calculation” sub-heading, the reviewer is concerned about the 20Å out-of-plane vacuum considered in this work, which might not be sufficient to eliminate the interactions from the out-of-plane periodic image completely. The authors need to justify this point.

R7: Thanks for the comments. Here, we have tested the vacuum thickness from 10 Å to 50 Å for the three phases, as shown in **Fig. R5**. The energies for prototype H and T phases converge to a certain value when the vacuum is 20 Å. For the Q phase, there is an energy difference of 2 meV between the 20 Å and 50 Å vacuum, which is quite tolerable compared to the relative energy between the three phases. To eliminate the interactions in periodic images, we reperformed all the calculations with 50 Å in the revised manuscript.

Fig. R5. The energy of the (a) prototype H, (b) T and (c) Q phase with vacuum thickness from 10 Å to 50 Å.

C8. The authors are requested to briefly discuss a few recent theoretical studies on GaSe in the “Introduction” section.

R8: We are very grateful for the reviewer’s constructive comment, we have added the following recent theoretical studies on GaSe in the "Introduction" section. “In recent years, GaSe has attracted much attention because it possesses many unique physical properties, such as topological insulator phase transition [*Phys. Rev. Lett.* **108**, 266805 (2012)], Lifshitz transition induced by sombrero-shape valence band [*Phys. Rev. B*

87,195403 (2013)], tunable parabolic to ring-shaped valence band [Phys. Rev. B **90**, 235302 (2014)], tunable ferromagnetism and half-metallicity [Phys. Rev. Lett. **114**, 236602 (2015)].”

Responses to Reviewer #2

C: In this manuscript, Li et al. reported signatures of room temperature ferroelectricity in GaSe flakes down to the monolayer limit. They claim the ferroelectricity comes from a new mechanism, i.e. intralayer sliding of the Se atomic sublayers. In the perspective of the reviewer, although the results appear interesting, the claim is still premature.

R: We thank the Reviewer for the insightful and constructive comments/suggestions on our manuscript. We have addressed all comments and made necessary revisions to the manuscript. The detailed point-by-point response is as follows.

C1. In several places (including in the title, Line 71-73 etc.), the authors refer monolayer or primitive unit cells of GaSe as “centrosymmetric”. In fact, only the primitive unit cell of β -GaSe (AA'AA'... stacking) is centrosymmetric, while the other three polytypes (including γ -GaSe) are not; monolayer GaSe (as monolayer MoS₂ in the 1H phase) is noncentrosymmetric. Maybe the authors mean nonferroelectric or nonpolar?

R1: Thanks for the constructive comment. Our previous considerations were not particularly adequate, indeed, GaSe monolayer has out-of-plane symmetry, i.e., mirror symmetry, rather than centrosymmetry. In the revised manuscript, we change the sentence “all the GaSe polytypes have the same primitive unit cells with a centrosymmetric structure” to “all the GaSe polytypes have the same primitive unit cells with a mirror symmetric structure in monolayer” Similarly, the related "centrosymmetry" in other places in the manuscript has been modified to "mirror symmetry". To be more rigorous, we changed the title of the manuscript to "Emergence of ferroelectricity in a non-ferroelectric monolayer", and this modification does not affect our main conclusions.

C2. The authors also mentioned interlayer sliding ferroelectricity in bilayer TMDs, bilayer hBN and 1T' ReS₂. As we know, in van der Waals materials adjacent layers are bonded via weak van der Waals forces; within the layer, it's covalent bond (much stronger). So, one would expect intralayer sliding to be much more difficult than interlayer sliding. The authors should provide theoretical calculations of energies of the different phases (Q , T and Q'), and check the feasibility of intralayer sliding and ferroelectric switching.

R2: Thanks for the suggestion. As shown in the TEM images in Fig. 2c of the manuscript, there are translations between sublayers in the GaSe monolayer. Thus, we performed Climbing Image Nudged Elastic Band (CI-NEB) method [*J. Chem. Phys.*

113, 9901-9904 (2000)] to calculate the energy evolution of GaSe monolayer between the ferroelectric Q/Q' phase and centrosymmetric T phase. As shown in **Fig. R6b**, the metastable Q/Q' is about 0.2 eV higher in energy than the T phase, and there is an energy barrier of 0.648 eV between them, while the coercive voltage is as large as ~ 6 V in the experiment. Such energy barrier between the Q and T phase can be comparable with other 2D ferroelectric materials in the same order, e.g., CuInP₂S₆ (0.15 eV, coercive voltage ~ 4 V) [*Nat. Mater.* **19**, 43-48(2020)].

Fig. R6. (a) The geometric structure of the Q phase. (b) The relative energy along the CI-NEB path between the Q/Q' and the T phase.

Moreover, this energy barrier can be largely reduced when layer thickness increases. As experimentally measured in the TEM images, the GaSe monolayer thickness is 0.53 ± 0.02 nm (Fig. 2c), which is larger than the calculated monolayer thickness (0.48 nm for T phase, 0.49 nm for Q phase). As shown in **Fig. R7(a)**, the evaluated energy barrier will decrease with the thickness increasing. When the thickness is 0.55 nm, the barrier can be reduced to 0.391 eV. Furthermore, considering that the experimental GaSe is p-type doped (Fig. 5d), the energy barrier is further reduced to 0.26 eV when increasing hole doping concentration (**Fig. R7(b)**). Therefore, it is easy to switch the polarization reversal by a moderate coercive voltage (~ 6 V) under the experimental conditions with thickness expansion and p-type doping.

Fig. R7. (a) Theoretically calculated reversal barrier of monolayer GaSe at different thicknesses. The gray area refers to the experimentally measured thickness of monolayer GaSe. (b) The reversal barrier under p-type doped GaSe condition, where the thickness of monolayer GaSe is taken as 0.55

nm.

Actually, the sliding mainly occurs between Ga and Se atoms, mainly stemming from the fact that Ga-Se bond is not a strong covalent bond within GaSe monolayer. As the electron localized function (ELF) [*Nature* **371**, 683–686 (1994)] shown in **Fig. R8**, the Ga-Se bond is more likely ionic bonding without preferred bond orientation, which is beneficial for the feasible polarization reversal in monolayer GaSe.

Fig. R8. The 2D electron localized function (ELF) of Q phase GaSe cut along Ga-Se bonds plane. The range of isosurface is from 0 to 0.65 e/Bohr³.

Revision: Based on the reviewer’s comment, we have added **Fig. R6b** and **Fig. R7** as Supplementary Fig. 5 and Supplementary Fig. 6 in the revised Supplementary Information, respectively. Besides, we have added the following discussion to the revised manuscript “When the Se atoms of upper sublayer move along the [110] direction, GaSe undergoes a centrosymmetric T phase. Then, the Se atoms of the down sublayer similarly move along the [110] direction and climb over an energy barrier of 0.648 eV to enter the Q’ phase (Supplementary Fig. 5), leading to the simultaneous reversal of IP and OOP polarization. Under experimental conditions with an extended range of sample thickness and p-type doping, the energy barrier will be further reduced to 0.26 eV (Supplementary Fig. 6), which can be overcome by applying a moderate coercive field (6V).”

C3. Thin GaSe flakes are known to be air sensitive (Arora & Erbe, InfoMat 3, 662, 2021), they can degrade fast in ambient conditions, giving unrealizable PFM and transport signals. Have the authors tried to passivate the GaSe flakes, especially for monolayer GaSe? How stable are the measurements, for example over a few hours?

R3: Thanks for the constructive comments. The oxidation process of GaSe is influenced by multiple factors, such as oxygen, humidity, and photo-induced [*Appl. Phys. Lett.* **17**, 173103 (2015)]. We prepared monolayer GaSe on HOPG substrate by MBE method (**Fig. R9a**), and investigated the oxidation process by XPS. **Figure R9(b, c)** shows the XPS spectra of monolayer GaSe exposure to air with evolution in time. After 25 h exposure to air, the Se 3d^{2/3} and Se 3d^{2/5} doublet peaks began to appear no longer obviously separated, and gradually merged into one peak. In addition, the positions of both Se 3d and Ga 3d peak shift towards lower binding energy, indicating the change

of the chemical valence state of Ga and Se, and the appearance of oxidation products, such as Ga₂O₃ and a-Se [*Semicond. Sci. Tech.* **32**, 105004 (2017)]. However, our PFM experiments on GaSe monolayer are performed on the sample exposure to air within one hour. Therefore, we believe the sample of GaSe monolayer does not degrade during PFM measurements.

Fig. R9. (a) Morphology of monolayer GaSe on HOPG substrate. (b, c) Evolution in time of XPS spectra of Ga 3d and Se 3d core levels during GaSe oxidation.

We also investigated exfoliated GaSe sample with a thickness about 3.6 nm (**Fig. R10(a, b)**) by Raman spectroscopy characterization. As shown in the **Fig. R10(c)**, The Raman spectra of few-layer GaSe show four Raman characteristic vibrational modes at 19.4 cm^{-1} (E_{2g}^2), 134.8 cm^{-1} (A_{1g}^1), 214 cm^{-1} (E_{2g}^1), 309 cm^{-1} (A_{1g}^2), while missing three Raman phonon modes appeared in thicker samples at 59.1 cm^{-1} (E_{1g}^1), 235 cm^{-1} ($A_{2}''(\text{TO})$), 247.5 cm^{-1} ($A_{2}''(\text{LO})$). Beyond that associated with GaSe, additional Raman modes were observed at 202 and 250 cm^{-1} . The broad peak near 250 cm^{-1} with a shoulder on the low-wavenumber side after 48 hours exposure to the air should be the Raman feature of a-Se [*J. Appl. Phys.* **107**, 073517(2010)], while the Raman peak at 202 cm^{-1} should be identified as the characteristic vibrational mode of Ga₂O₃ [*J. Cryst. Growth.* **401**, 330-333 (2014)]. At room temperature, the main products of GaSe oxidation are Se and Ga₂O₃ [*Nanotechnology.* **28**, 175701(2017)]. Moreover, with a slight increase in excitation light power, GaSe oxidizes rapidly in a few minutes, indicating that it is highly susceptible to photoinduced oxidation, as shown in **Fig. R10(d)**. Therefore, prolonged light tests or strong light tests can lead to GaSe oxidation quickly, and the surface after photo-induced oxidation leads to the particle cluster morphology occurring, as shown in the area marked by small circles in **Fig. R10(a, b)**.

Fig. R10. (a, b) The Optical micrograph and AFM morphology of exfoliated GaSe sample with a thickness of about 3.6 nm, respectively. The oxidized area after Raman light illumination is marked by small circles. (c) Evolution in time of Raman spectra of GaSe, tested with 532 nm laser and power density of $0.05 \text{ mW } \mu\text{m}^{-2}$. (d) Raman spectra of GaSe at a power density of 0.05, 0.21, and $0.3 \text{ mW } \mu\text{m}^{-2}$, respectively. (e) Peak fitting of Raman spectra of oxidized GaSe. The short dash line represents the total fit of the data, additional peaks appear at 202 and 250 cm^{-1} identified as Ga_2O_3 and a-Se, respectively.

The optical techniques for characterizing the GaSe oxidation such as PL, XPS and Raman, can accelerate the oxidation by photo-inducement [*Appl. Phys. Lett.* **17**, 173103 (2015)]. However, our PFM experiments on GaSe sample are performed in a dry environment, and no laser introduced, in which the sample cannot be much degraded within 25 hours. Therefore, our experimental measurements should reflect the pristine properties of GaSe.

Revision: Based on the reviewer's comment, we have added **Fig. R9** and **Fig. R10** as Supplementary Fig. 17 and Supplementary Fig. 18 in the revised Supplementary Information, respectively. Besides, we have added the above-related discussion as Supplementary Note 18 in the revised Supplementary Information.

C4. Following the above question, regarding transport measurements in Figure 5, one needs to be extremely careful in distinguishing ferroelectric hysteresis from charging effect that could possibly come from defects and impurities. Have the authors tried to sweep the gate/bias at different rates? How the HRS evolve as a function of time (retention time of the "ferroelectricity")?

R4: Thanks for the comments. We believe that the hysteresis observed in Fig. 5(d,g) is mainly a result of the ferroelectricity of GaSe, because other non-ferroelectric FETs with the same device fabrication process were found to have almost no hysteresis [*Nat. Commun.* **10**, 1-8 (2019)]. In fact, the defects and impurities introduced during the device fabrication process may induce hysteresis when testing the transfer and output

curves of the device, and it's also very hard to distinguish them simply based on electrical tests. For example, a positively shifted threshold voltage can be observed at sweeping from 80 V to -80 V compared to sweeping from -80 V to 80 V (Fig. 5d), which may either result from the accumulated negative trapped charges at trap states in the dielectric layer SiO₂ or the SiO₂/GaSe interface [*Nat. Commun.* **10**, 1-8 (2019)], or result from the negative polarization bound charges near the bottom surface of the channel GaSe [*Adv. Funct. Mater.* **32**, 2205468 (2022)].

Nevertheless, defects and impurities were significantly decreased owing to the careful device fabrication process. To eliminate possible adsorbates on the surface SiO₂ that might induce unintended interface trap states, the SiO₂/Si substrate was immersed and heated at 120 °C for 3 hours in a mixture of concentrated sulfuric acid and hydrogen peroxide (mass ratio 3:1). And soon after that, the GaSe flake was directly exfoliated onto the SiO₂/Si substrate before source and drain electrodes were deposited. On the other hand, as evidence of the origin of the hysteresis window, the strong ferroelectricity of the GaSe was already shown in PFM tests in Fig. 3(b), in which the PFM amplitude and phase signals as functions of bias voltage exhibited a typical butterfly-like shape and a hysteresis loop with a 180° phase switching, respectively.

According to the reviewer's suggestion, we performed sweeping rate-dependent electrical tests shown in **Fig. R11**. Due to the charge trapping effect, the hysteresis loop window typically should become larger when the voltage sweep is slower [*Nat. Commun.* **13**, 7696 (2022); *IEEE. J. Electron. Devi.* **7**, 855-862 (2019)]. However, it can be seen in our test results that for both the transfer and output curves, the hysteresis loops window does not shift obviously during the voltage sweeping process, thus ruling out the effect of trap states.

Fig. R11. The sweeping rate-dependent (a) transfer at $V_{DS} = 10$ V and (b) output curves at $V_{GS} = 0$ V.

According to the reviewer's concerns about the retention time of the HRS, we have performed a long-term retention test on the HRS, as shown in **Fig. R12**. The HRS was stable for over 100 s, and showed almost no current change during this period.

Fig. R12. Retention performance of high resistance state (HRS) for over 100 s. A drain voltage of 10 V has been applied ahead for 2 s to set the HRS.

Revision: Based on the reviewer’s comment, we have added **Fig. R11** and **Fig. R12** as Supplementary Fig. 10 and Supplementary Fig. 12 in the revised Supplementary Information, respectively. Besides, we have added the following discussion to the revised manuscript: “The effect of trap states can be excluded from the unchanged hysteresis loops windows at different sweep rates (Supplementary Fig. 10), since the hysteresis loops windows induced by the trap states usually become larger when slowing down the voltage sweep rate”, as well as “The retention performance of HRS for over 100 s was also carried out and shown in Supplementary Fig. 12. The HRS was stable for over 100 s and showed almost no current change during this period.”

C5. Voltage pulses and current plateaus in Supplementary Figure 7 do not look synchronized. Could the authors clarify the behaviors together with IV and transfer characteristics?

R5: Thanks for the comment and suggestion. The unsynchronized voltage pulses and current plateaus in Supplementary Fig. 7 of the original Supplementary Information were induced by the sparse sampling points during the voltage withdrawing period due to an unintended improper parameter set of the semiconductor analyzer. To avoid ambiguity of the data, we have conducted the test again and narrowed the interval of sampling points, so that the voltage pulses and current plateaus are synchronized, as shown in **Fig. R13**.

According to the reviewer’s suggestion, a detailed description and internal mechanism of the HRS-LRS switching behaviors are illustrated. As shown in **Fig. R11(a)**, the channel resistance was programmed through drain bias voltage owing to the in-plane ferroelectricity of the channel GaSe. It can be seen in Fig. 5d that different bias values correspond to different drain currents, i.e., the HRS and LRS respectively. For out-of-plane ferroelectric polarization, after a gate pulse of 40 V was applied for 2 seconds and then withdrawn, the device was subsequently set to the LRS. On the contrary, after a gate pulse of -40 V was applied for 2 seconds and then withdrawn, the device was set to the HRS. This phenomenon is consistent with the output curve under different gate poling in Fig. 5g. When the positive gate voltage is removed, the output current is higher than the original value, and vice versa.

Based on the reviewer's comments, we have added **Fig. R13** in the revised Supplementary Information.

Fig. R13. The repeatability of ferroelectric resistance switching. (a) The resistance switching by applying the channel bias. (b) The resistance switching by applying the gate voltage.

C6. The authors claimed IP polarization signal is stronger than OOP polarization (Line 190) by comparing amplitudes of the lateral and vertical PFM. This is an unfair comparison since their responsivities could be rather different.

R6: Thanks for the comment. It is not quite appropriate to compare the magnitude of IP and OOP polarization only by comparing amplitudes of the lateral and vertical PFM, because their responsivities are different. Therefore, we have removed this statement “We found the IP polarization signal is relatively stronger than OOP polarization.” in the revised manuscript.

REVIEWER COMMENTS

Reviewer #1 (Remarks to the Author):

The authors have satisfactorily addressed all the concerns raised by the reviewer. Therefore, the manuscript may be accepted for publication.

Reviewer #2 (Remarks to the Author):

In the rebuttal, the authors resolved some of my concerns. However, the claim that the observed ferroelectric behaviors originate from intralayer sliding is still not conclusive/convincing to me. To eliminate the ambiguities, the authors are suggested to perform in-situ STEM before and after the ferroelectric switching. On the other hand, in their calculations (Fig. S5, Fig. R6b), the free energy of T phase (polarization $P=0$) is 0.2 eV lower than that of the metastable Q/Q' phase ($P\neq 0$). If this is the case, there would be no spontaneous phase transition, and we should see T phase GaSe in STEM.

The authors also mentioned the barrier of the ferroelectric switching can be lowered to 0.26 eV for hole doping with a concentration of 0.15 e/u.c. Have they check if this is reasonable in reality?

Responses to Reviewer #1

C: The authors have satisfactorily addressed all the concerns raised by the reviewer. Therefore, the manuscript may be accepted for publication.

R1: We thank the reviewer for the supporting on our manuscript.

Responses to Reviewer #2

CI: In the rebuttal, the authors resolved some of my concerns. However, the claim that the observed ferroelectric behaviors originate from intralayer sliding is still not conclusive/convincing to me. To eliminate the ambiguities, the authors are suggested to perform in-situ STEM before and after the ferroelectric switching. On the other hand, in their calculations (Fig. S5, Fig. R6b), the free energy of T phase (polarization $P=0$) is 0.2 eV lower than that of the metastable Q/Q' phase ($P\neq 0$). If this is the case, there would be no spontaneous phase transition, and we should see T phase GaSe in STEM.

R1: Thanks for the constructive comments.

As far as we know, no publications have reported observing polarization switching in 2D materials using *in-situ* STM. We attempted *in-situ* STEM characterization before and after ferroelectric switching by applying an electric field. However, characterizing 2D ferroelectric polarization switching using *in-situ* STEM is challenging for several reasons. Firstly, unlike insulating 3D oxide ferroelectrics, 2D ferroelectrics are generally semiconductors with lower resistance, which can cause current saturation and severe heating. This prevents us from applying sufficient voltage to switch the polarization. Secondly, applying voltage for *in-situ* STEM observation causes the sample to jitter severely, and the view area drifts, making it difficult to obtain clear images.

Nevertheless, we performed *ex-situ* STEM experiments before and after ferroelectric switching, and the results can confirm our conclusion that the ferroelectric behavior of GaSe originates from intralayer sliding. Before ferroelectric switching, we observed two distinct structures in different regions of the GaSe sample using STEM: the distorted Q phase (**Fig. R1a**) and the non-distorted H phase (*not T phase*) (**Fig. R1b**). Subsequently, we polarized the GaSe sample at -50 V for 1 minute and performed a STEM test again. The sample was switched to the distorted Q' phase with an opposite distortion direction compared to the Q phase (**Fig. R1c**). Therefore, we speculate that the ferroelectric Q' phase is generated by the intralayer sliding and polarization flip of the Q phase.

Next, we performed theoretical calculations to verify this hypothesis. Based on the STEM images of distorted GaSe before and after ferroelectric switching, we construct the ferroelectric Q and Q' phase in DFT calculations, as shown in **Fig. R2**. The reversal process of polarization is analyzed by inserting points between the Q and Q' structures, and each inserted structure is fully optimized by fixing in-plane coordinates while relaxing out-of-plane coordinates. As the energy variation during polarization reversal process is shown in **Fig. R3a**, the Se atoms of the upper sublayer move along [110]

direction first overcoming a small barrier of 0.04 eV/u.c. Then, the GaSe monolayer experiences a stable *H phase* which is 0.68 eV lower in energy than the metastable Q phase. After that, a large electric field may assist a portion of Se atoms in the bottom sublayers move along [110] direction, climbing over a barrier of 0.72 eV/u.c. to the metastable Q' phase. As the OOP polarization variation shown in **Fig. R3b**, the calculated OOP polarization of Q phase is -4.89 pC/m and polarization of Q' phase is reversed to 4.89 pC/m. Therefore, due to the barrier and large electric field, we can observe the coexistence of *H phase* and Q phase in the STEM images.

To conclude, the observed ferroelectric behaviors in GaSe originate from intralayer sliding. To the best of our knowledge, this is perhaps the first work in the field of 2D ferroelectrics to directly observe 2D ferroelectric polarization switching by STEM.

Fig. R1 STEM images of distorted Q phase (a), non-distorted H phase (b) and distorted Q' phase (c) of GaSe. The distorted Q' phase is generated by intralayer sliding of the Q phase after the applied electric field polarization. The insets are enlarged images, respectively.

Fig. R2 The geometric structure of the Q, H and Q' phases GaSe monolayer. The orange, dark green and light green balls denote Ga atoms and Se atoms of the upper sublayer and bottom sublayer. The red and blue arrows represent the direction of polarization.

Fig. R3 (a) The relative energy change of the ferroelectric phase transition process from Q to H to Q' phase. The energy of H phase is set to zero. (b) The variation of OOP polarization is described by Se atom displacement, and the zero displacement is defined as the H phase.

Revision: Based on the reviewer's comment, we have added **Fig. R1** as Fig. 4(a-c), **Fig. R2** as Fig. 4d, **Fig. R3b** as Fig. 4g, as well as the related discussion in the revised manuscript. All amendments in the manuscript have been highlighted in yellow.

C2: The authors also mentioned the barrier of the ferroelectric switching can be lowered to 0.26 eV for hole doping with a concentration of 0.15 e/u.c. Have they check if this is reasonable in reality?

R2: Thanks for the constructive comments.

The hole concentration of 0.15 e/u.c. is ideal in DFT calculation, to illustrate that p-type doping condition is beneficial for reducing barrier. Theoretical work simulated the hole doping concentration of GaSe can reach this order of magnitude (*Phys. Rev. Lett.* **114**(23), 236602 (2015)), but it is still difficult to achieve experimentally. To avoid misunderstanding, we revised the statement in the main text. “Under experimental conditions with an extended range of sample thickness and p-type doping, the energy barrier will be further reduced to 0.26 eV (Supplementary Fig. 6)” is revised to “Under experimental conditions with extended sample thickness and p-type doping, the energy barrier will be further reduced (Supplementary Fig. 6)”. Here we reconsider smaller p-type doping concentration achievable in the experiment and the barrier can be reduced by 0.14 eV as in **Fig. R4** and the polarization reversal barrier is of the same order of magnitude as other ferroelectric materials. Besides, the external electric field (6V) is large enough to overcome the barrier as we observe the changes of GaSe structure after applying electric field in the STEM images.

Fig. R4. The relative energy change of the ferroelectric phase transition process from Q to H to Q' phase. The thickness is taken as 0.51 nm and the hole doping concentration is 0.05 hole/u.c.

REVIEWER COMMENTS

Reviewer #2 (Remarks to the Author):

The reviewer is happy about the appended STEM measurements and the revised discussion on electrostatic doping. But the free energy - Se displacement (electric polarization) profile in Fig. R3a is still confusing. Let me be explicit about my questions. In the classical Ginzburg-Landau theory for ferroelectric phase transitions, below the critical temperature, a double-well energy landscape is expected, accompanied by a spontaneous phase transition. If the energy landscape for GaSe is as presented (Fig. R3a), the Q and Q' phases are metastable, there should be no spontaneous phase transition from the highly symmetric phases (could be in the H phase at temperatures above the critical temperature). And the electric polarization observed at room temperature may be a consequence of built-in electric or strain field, which could be nonuniform (this may explain the coexistence of Q and H phases in the original STEM images). If this is the case, the ferroelectricity reported in this work is not intrinsic (at least not spontaneous).

Responses to Reviewer #2

*C1: The reviewer is happy about the appended STEM measurements and the revised discussion on electrostatic doping. But the free energy-*Se* displacement (electric polarization) profile in Fig. R3a is still confusing. Let me be explicit about my questions. In the classical Ginzburg-Landau theory for ferroelectric phase transitions, below the critical temperature, a double-well energy landscape is expected, accompanied by a spontaneous phase transition. If the energy landscape for GaSe is as presented (Fig. R3a), the *Q* and *Q'* phases are metastable, there should be no spontaneous phase transition from the highly symmetric phases (could be in the *H* phase at temperatures above the critical temperature). And the electric polarization observed at room temperature may be a consequence of built-in electric or strain field, which could be nonuniform (this may explain the coexistence of *Q* and *H* phases in the original STEM images). If this is the case, the ferroelectricity reported in this work is not intrinsic (at least not spontaneous).*

R1: Thanks for the constructive comments, which prompt us to explore more deeply the origin of GaSe ferroelectricity. Previous theoretical work proposed that the out-of-plane polarization in *h*-NbN originates from the coupling of strain and electric field, and that the ferroelectric phase transition process uses a triple-well potential landscape [*Phys. Rev. Lett.* 123, 037601 (2019)]. The unconventional ferroelectricity of this mechanism is relatively rare. In our work, the paraelectric *H* phase is 0.68 eV lower than the ferroelectric *Q* phase, and the *Q* and *Q'* phases are metastable. Therefore, we agree with the reviewer that the ferroelectricity reported in our work is not spontaneous. The observed polarization may be the result of a built-in electric or strain field, which is not nonuniform, leading to the coexistence of the *Q* and *H* phases in the original STEM images. When the reversing electric field is applied, the *Se* atoms in the upper layer of *Q* phase first move back to the *H* phase without OOP polarization. Then, a portion of the *Se* atoms in the bottom sublayer move to the *Q'* phase, and the OOP polarization is reversed to 4.89 pC/m pointing upward. Therefore, we have modified the statement in the revised manuscript that the polarization is not spontaneous, and the built-in electric or strain field may promote the appearance of the distorted *Q* phase. All amendments in the manuscript have been highlighted in yellow.

REVIEWERS' COMMENTS

Reviewer #2 (Remarks to the Author):

The authors resolved all my concerns, the paper may be published in Nature Communications.

Reviewer #2:

C: The authors resolved all my concerns, the paper may be published in Nature Communications.

R: Thanks for reviewer's supporting on our manuscript.